# Efficient Multi-modal Dataset Distillation via Analytic Parameter Matching

**Deyu Bo** [1]    **Xinchao Wang** [1]

## Abstract

Multi-modal dataset distillation (MDD) seeks to compress large-scale multi-modal datasets into a compact set of synthetic pairs. Existing methods employ a dual-trajectory matching framework to align the teacher and student models within each modality. While effective, this paradigm incurs non-negligible memory and computational overhead due to the checkpoint storage and bi-level optimization over synthetic data. To address these limitations, we propose analytic parameter matching (APM), which theoretically derives the analytic parameters of modal projectors to replace the inner-loop optimization, and then aligns the analytic projector parameters of teacher and student models. APM offers two key advantages: (1) it replaces checkpoint-intensive storage with only two cached matrices, significantly reducing memory consumption; and (2) it computes analytic parameters in a single forward pass, thereby avoiding costly bi-level optimization. Empirically, APM achieves up to 65× storage reduction and 9.6× faster distillation, while scaling to 1,000 synthetic pairs. Extensive experiments on image-text and audio-text benchmarks demonstrate the effectiveness of APM in cross-modal retrieval tasks, e.g., 12.8 IR@1 and 17.8 TR@1 in Flickr30k with 100 synthetic pairs. Moreover, APM exhibits notable generalization performance in cross-architecture evaluation and zero-shot classification tasks.

## 1. Introduction

Dataset distillation (DD) (Wang et al., 2018) has emerged as a *de facto* framework for improving data efficiency and accelerating the training of neural networks (Yu et al., 2024; Lei & Tao, 2024). Traditional DD methods focus on compressing the large-scale vision datasets, *e.g.*, CI-

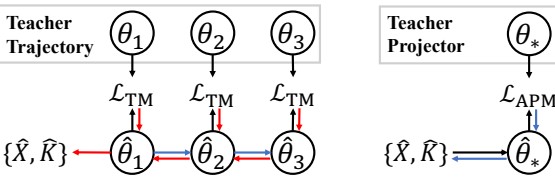

*(a)* TM: **Optimized** Parameters   *(b)* APM: **Analytic** Parameters

*Figure 1.* Comparison between Trajectory Matching (TM) and Analytic Parameter Matching (APM). Instead of optimizing parameters via gradient descent, APM directly calculates the analytic parameters of modal projectors, reducing the time and space overhead. Forward (→), First Backward (→), Second Backward (→).

*Table 1.* The storage, time, and space overhead of three MDD methods. APM has a 65× storage reduction and 9.6× speedup over LoRS (Xu et al., 2024a).

| Method | Buffer (Offline) | Distillation (Online) | |
| --- | --- | --- | --- |
| | Storage (GB) | Time (s/iter) | Space (GB) |
| LoRS | 32.6 | 11.50 | 21.78 |
| RepBlend | 14.6 | 1.71 | 10.17 |
| APM | 0.5 | 1.20 | 11.17 |

FAR (Krizhevsky et al., 2009) and ImageNet-1k (Deng et al., 2009) into smaller yet representative ones. Roughly speaking, these methods can be divided into three categories: Gradient Matching (Zhao et al., 2021; Kim et al., 2022; Liu et al., 2023), Trajectory Matching (Cazenavette et al., 2022; Guo et al., 2024), and Statistical Matching (Zhao & Bilen, 2023; Yin et al., 2023; Shao et al., 2024). Recently, the distillation of multi-modal datasets (Wu et al., 2024), *e.g.*, images and text, has drawn increasing attention due to its broader applications in downstream tasks such as cross-modal retrieval and conditional generation.

Existing multi-modal dataset distillation (MDD) methods (Wu et al., 2024; Xu et al., 2024a; Zhang et al., 2025; Dang et al., 2025) adopt trajectory matching (TM) as the distillation framework, where expert trajectories are used to supervise the student models trained on the synthetic dataset. Despite its effectiveness, this framework suffers from two visible drawbacks: First, TM requires storing the entire teacher trajectories, *e.g.*, a series of checkpoints $\{\theta_1, \theta_2, \theta_3\}$ in Figure 1a, leading to significant storage overhead. For example, LoRS (Xu et al., 2024a) trains 20 trajectories, each

[1]National University of Singapore. Correspondence to: Xinchao Wang <xinchao@nus.edu.sg>.

*Proceedings of the 43rd International Conference on Machine Learning*, Seoul, South Korea. PMLR 306, 2026. Copyright 2026 by the author(s).

containing 10 model checkpoints. This takes up over 30GB of space, even larger than the dataset itself, as shown in Table 1. Second, TM involves bi-level optimization during distillation, which first updates the model parameters and then optimizes the synthetic dataset by minimizing the differences between teacher and student trajectories, limiting its efficiency and scalability, as shown in Figure 1a.

Once the weaknesses of existing MDD methods are identified, it is natural to ask: *How can we improve the efficiency and scalability of MDD while preserving its effectiveness?* To answer this question, we first note that the computational bottleneck of MDD stems from the inner-loop optimization over synthetic datasets. Instead of relying solely on the gradient descent, a more efficient alternative is to explore its analytic formulations. However, this is a non-trivial task, as neural networks contain numerous nonlinear functions, which prevent the computation of analytic parameters.

As the first contribution of this paper, we derive the analytic parameters of the linear and nonlinear projectors under the InfoNCE (van den Oord et al., 2018) loss with softmax activation. Based on this insight, we propose analytic parameter matching (APM), as shown in Figure 1b, which first computes the analytic parameters of modal projectors on the real and synthetic datasets, and then minimizes their discrepancy to optimize the synthetic dataset. Instead of storing the entire trajectory, APM only caches the analytic parameters of real modal projectors, which significantly reduces the storage budget, as shown in Table 1. Furthermore, APM eliminates bi-level optimization, as the analytic parameters of synthetic modal projectors can be obtained in the forward pass, thereby further improving its efficiency and scalability. The contributions of this paper are summarized below:

- We analyze the limitations of existing MDD methods, highlighting their substantial storage overhead due to storing multiple model trajectories and their inefficiency caused by bi-level optimization.

- We propose APM, which replaces inner-loop optimization with analytic parameters of modal projectors, removing the need for trajectory storage and bi-level optimization. This paradigm yields up to $65\times$ reduction in memory usage and $9.6\times$ improvement in distillation efficiency.

- Extensive experiments on the image-text and audio-text datasets demonstrate that APM achieves superior performance over state-of-the-art MDD methods and maintains strong generalization across downstream settings, including zero-shot classification.

## 2. Preliminaries

Before presenting our method, we introduce some key concepts relevant to this work. More detailed discussions can be found in Section 5.

**Multi-modal Contrastive Learning (MCL)** aims to learn a shared embedding space across modalities, where semantically matched samples, *e.g.*, an image and its caption, are pulled together, while unmatched samples are pushed apart. Consider an image–text dataset with paired samples $(\boldsymbol{x}_i, \boldsymbol{\kappa}_i) \in \mathcal{D}$, where $\boldsymbol{x}_i$ represents the $i$-th image, and $\boldsymbol{\kappa}_i$ denotes its caption. To project data into the shared space, MCL trains a vision–language model $\mathcal{M} = \{f_\mathrm{E}, f_\mathrm{P}, g_\mathrm{E}, g_\mathrm{P}\}$, where $f_\mathrm{E}$ and $f_\mathrm{P}$ denote the image encoder and projector, and $g_\mathrm{E}$ and $g_\mathrm{P}$ are the text encoder and projector, respectively. Finally, a contrastive learning loss is adopted to optimize the model. This learning process can be described as:

$$\boldsymbol{u}_i = \frac{f_\mathrm{P}\big(f_\mathrm{E}(\boldsymbol{x}_i)\big)}{\big\|f_\mathrm{P}\big(f_\mathrm{E}(\boldsymbol{x}_i)\big)\big\|_2}, \quad \boldsymbol{v}_i = \frac{g_\mathrm{P}\big(g_\mathrm{E}(\boldsymbol{\kappa}_i)\big)}{\big\|g_\mathrm{P}\big(g_\mathrm{E}(\boldsymbol{\kappa}_i)\big)\big\|_2},$$

$$\mathcal{L}_\mathrm{NCE} = -\frac{1}{|\mathcal{D}|} \sum_{i=1}^{|\mathcal{D}|} \log \frac{\exp(z_{ii})}{\sum_{j=1}^{|\mathcal{D}|} \exp(z_{ij})}, \quad (1)$$

where $z_{ij} = \boldsymbol{u}_i^\top \boldsymbol{v}_j / \tau$ measures the similarity between image and text, and $\tau$ is a temperature ratio. By narrowing the gap between positive pairs and enlarging the gap between negative pairs, $\mathcal{M}$ can learn the semantic correspondence between images and text, which can be used in downstream retrieval or generation tasks.

**Multi-modal Dataset Distillation** seeks to learn a compact set of synthetic pairs $\mathcal{S} = \{(\hat{\boldsymbol{x}}_i, \hat{\boldsymbol{\kappa}}_i)\}_{i=1}^{|\mathcal{S}|}$, where $|\mathcal{S}| \ll |\mathcal{D}|$, such that a multi-modal model trained on $\mathcal{D}$ and $\mathcal{S}$ will have comparable performance. We formulate this task as a bi-level optimization problem:

$$\min_{\mathcal{S}} \sum_{j=1}^{|D|} \mathcal{L}_\mathrm{NCE}(\mathcal{M}^*(\boldsymbol{x}_j, \boldsymbol{\kappa}_j)), \quad (2)$$

$$\mathcal{M}^* = \arg\min_{\theta} \sum_{i=1}^{|S|} \mathcal{L}_\mathrm{NCE}(\mathcal{M}(\hat{\boldsymbol{x}}_i, \hat{\boldsymbol{\kappa}}_i)),$$

where the inner loop trains the model $\mathcal{M}$ on the synthetic data until convergence, and the outer loop optimizes the synthetic data by minimizing the loss function on the real data. However, solving this bi-level optimization issue is time-consuming. Existing methods (Wu et al., 2024; Xu et al., 2024a) adopt the trajectory matching (Cazenavette et al., 2022) as a surrogate, which minimizes the model optimization trajectories between the real and synthetic data. Despite some efforts, TM requires bi-level optimization during training, which greatly limits its efficiency and scalability. This observation motivates the design of our model.

## 3. The Proposed Method

In this section, we introduce our proposed method in detail. We begin by deriving the analytic parameters for the image and text projectors, followed by the formulation of the objective function for APM. The overall pipeline of APM is illustrated in Figure 2.

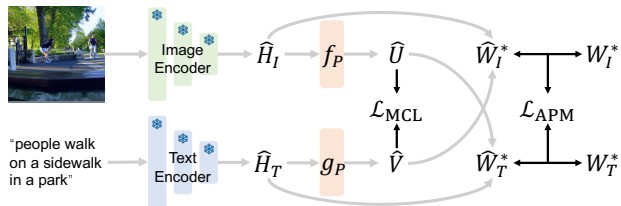

*Figure 2.* Pipeline of APM. We use the gray arrow to represent the forward pass and the black arrow to denote the calculation of loss functions. Here, $\hat{W}_I^*$ and $\hat{W}_T^*$ are the analytic parameters of the synthetic datasets, and the analytic parameters of real datasets, *i.e.*, $W_I^*$ and $W_T^*$, are pre-calculated.

## 3.1. Analytic Solutions of Modal Projectors

To improve the efficiency of MDD, we propose to align the analytic parameters of the modal projectors on real and synthetic datasets, rather than matching their trajectories. The advantages are two-fold: First, we can throw off the massive model checkpoints and focus on distilling the essentials of the dataset. Second, we avoid bi-level optimization and only need to propagate the gradient once, which significantly improves the efficiency and scalability of MDD.

However, it is hard to calculate the analytic parameters for all parameters due to the non-linearity of neural networks. To solve this issue, we switch to match the image and text projectors as they carry the semantic information across modalities. Specifically, we study the CLIP-style (Radford et al., 2021) network architecture, containing two linear projectors, *i.e.*, $f_P = W_I$ and $g_P = W_T$. For clarity, we use the matrix form to represent the set of $\{x_i\}_{i=1}^{|\mathcal{D}|}$ and $\{\kappa_i\}_{i=1}^{|\mathcal{D}|}$, denoted as $X$ and $K$. As a result, Equation 2 can be reformulated as:

$$H_I = f_E(X) \in \mathbb{R}^{|\mathcal{D}| \times d_I}, \quad H_T = g_E(K) \in \mathbb{R}^{|\mathcal{D}| \times d_T},$$
$$U = H_I W_I \in \mathbb{R}^{|\mathcal{D}| \times d}, \quad V = H_T W_T \in \mathbb{R}^{|\mathcal{D}| \times d},$$

where $d_I$ and $d_T$ are the embedding dimensions of image and text, respectively, and $d$ is the dimension of the shared semantic space. Then we have the following propositions:

**Proposition 3.1.** *For the linear projectors $U = H_I W_I$ and $V = H_T W_T$, $\mathcal{L}_{NCE}$ has analytical solutions with respect to $W_I$ and $W_T$, defined as:*

$$W_I^* = \frac{|\mathcal{D}|}{2\tau} \underbrace{\left(H_I^\top H_I\right)^{-1} H_I^\top}_{\textit{Image}} \underbrace{V \left(V^\top V\right)^{-1}}_{\textit{Text}},$$

$$W_T^* = \frac{|\mathcal{D}|}{2\tau} \underbrace{\left(H_T^\top H_T\right)^{-1} H_T^\top}_{\textit{Text}} \underbrace{U \left(U^\top U\right)^{-1}}_{\textit{Image}}. \quad (3)$$

*Proof.* See Appendix A.1. □

Although linear projectors are widely used in the CLIP-style architectures, the non-linear projectors have better expres-

sive power. Therefore, we further extend the analytical parameters to the case of nonlinear projectors:

**Proposition 3.2.** *For non-linear projectors $U = \sigma(H_I W_I)$ and $V = \sigma(H_T W_T)$, the analytic solutions becomes:*

$$W_I^* = (H_I^\top H_I)^{-1} H_I^\top \sigma^{-1}\left(\frac{|\mathcal{D}|}{2\tau} V (V^\top V)^{-1}\right), \quad (4)$$

$$W_T^* = (H_T^\top H_T)^{-1} H_T^\top \sigma^{-1}\left(\frac{|\mathcal{D}|}{2\tau} U (U^\top U)^{-1}\right), \quad (5)$$

*where $\sigma^{-1}(\cdot)$ is the inverse function of the activation function $\sigma(\cdot)$.*

*Proof.* See Appendix A.2. □

We can observe that the linear and non-linear projectors have similar analytic parameters. Considering that some activations, such as ReLU, do not have inverse functions, we will use the linear projectors for the subsequent analysis. Building on Proposition 3.1, the optimal projector for each modality can be decomposed into two important factors: image and text representations. Here, we take $W_I^*$ as an example, explain the benefit for MDD, and interpret its underlying intuition.

**Data Entropy.** Real-world multi-modal datasets typically have low-rank structures (Lee & Chung, 2025), where a few singular values dominate the data. The analytic parameter $W_I^*$ inverses the singular values to increase the rank of data representations: $(H_I^\top H_I)^{-1} H_I = U S^{-1} V$, where $H_I = U S V$ denotes the SVD decomposition. By matching the analytic parameters of real and synthetic datasets, APM can produce synthetic data with higher data entropy, thereby encoding more knowledge from real datasets. See Figure 3 for an experimental validation.

Moreover, a closely related work, CovMatch (Lee & Chung, 2025), also explores the matching of cross-covariance matrices, *i.e.*, $U^\top V$, in MDD. However, it only considers the inter-modal information, while the intra-modal information, *e.g.*, $(H_I^\top H_I)^{-1}$, is overlooked, leading to suboptimal results. See Tables 2 and 3 for the performance comparison.

## 3.2. Analytic Parameter Matching

Once the advantages of analytic modal projectors are identified, the next step is to align the distributions of the real and synthetic datasets by matching their analytic parameters. However, there are some instabilities in the calculation of Equation 3: (1) **Embedding Shift.** Matrix whitening requires embeddings to have zero mean (Kessy et al., 2018), but the analytic parameters omit it, which may result in embedding shift. (2) **Scale Explosion.** The whitening matrix involves the sum of sample outer products, *i.e.*, $\left(H_I^\top H_I\right)^{-1/2} = \left(\sum_i h_i^\top h_i\right)^{-1/2}$, which may affect

the scale of the analytic parameters of real and synthetic datasets. (3) **Matrix Inversion.** As the size of the synthetic dataset is less than the embedding dimension, the analytic parameters of the synthetic dataset are not full-rank[1]. As a result, directly calculating its inversion may lead to unstable distillation.

To overcome these issues, we reformulate the analysis parameters into three parts: image term, cross term, and text term, and introduce three corresponding normalizations to stabilize the distillation process:

$$H_I^\top H_I \approx \Sigma_{II} = \frac{1}{|\mathcal{D}|}(H_I - \boldsymbol{\mu}_I)^\top (H_I - \boldsymbol{\mu}_I) + \alpha I,$$

$$H_I^\top V \approx \Sigma_{IV} = \frac{1}{|\mathcal{D}|}(H_I - \boldsymbol{\mu}_I)^\top (V - \boldsymbol{\mu}_V),$$

$$V^\top V \approx \Sigma_{VV} = \frac{1}{|\mathcal{D}|}(V - \boldsymbol{\mu}_V)^\top (V - \boldsymbol{\mu}_V) + \alpha I,$$

$$H_T^\top H_T \approx \Sigma_{TT} = \frac{1}{|\mathcal{D}|}(H_T - \boldsymbol{\mu}_T)^\top (H_T - \boldsymbol{\mu}_T) + \alpha I,$$

$$H_T^\top U \approx \Sigma_{TU} = \frac{1}{|\mathcal{D}|}(H_T - \boldsymbol{\mu}_T)^\top (U - \boldsymbol{\mu}_U),$$

$$U^\top U \approx \Sigma_{UU} = \frac{1}{|\mathcal{D}|}(U - \boldsymbol{\mu}_U)^\top (U - \boldsymbol{\mu}_U) + \alpha I.$$

where $\boldsymbol{\mu}_I$, $\boldsymbol{\mu}_T$, $\boldsymbol{\mu}_U$, and $\boldsymbol{\mu}_V$ denote the mean values of $H_I$, $H_T$, $U$, and $V$, respectively. The hyperparameter $\alpha$ is used to ensure that the matrix is full-rank. Besides, for the synthetic dataset, we use a hat notation to represent their corresponding analytic parameters, *e.g.*, $\hat{\Sigma}_{II}$, and do not repeat their definitions for clarity. The objective function of APM is defined as:

$$\begin{aligned}
\mathcal{L}_{\text{APM}} &= ||W_I^* - \hat{W}_I^*||_F^2 + ||W_T^* - \hat{W}_T^*||_F^2 \\
&= ||\Sigma_{II}^{-1}\Sigma_{IV}\Sigma_{VV}^{-1} - \hat{\Sigma}_{II}^{-1}\hat{\Sigma}_{IV}\hat{\Sigma}_{VV}^{-1}||_F^2 \\
&\quad + ||\Sigma_{TT}^{-1}\Sigma_{TU}\Sigma_{UU}^{-1} - \hat{\Sigma}_{TT}^{-1}\hat{\Sigma}_{TU}\hat{\Sigma}_{UU}^{-1}||_F^2. \quad (6)
\end{aligned}$$

It is worth noting that calculating the analytic parameters will introduce quadratic complexity with respect to the number of samples. To avoid this issue, we need to pre-compute the analytic parameters of the real dataset to reduce the time and space overhead during distillation. Therefore, we first pre-train a teacher model $\mathcal{M}^t$ on the real dataset and freeze its weights during the distillation process, so that we can cache the analytic parameters based on the fixed modal encoders and projectors. Finally, we combine the objectives of MCL and APM as the overall loss function for distillation:

$$\mathcal{L} = \sum_{i=1}^{|\mathcal{S}|} \mathcal{L}_{\text{NCE}}(\mathcal{M}^t(\hat{\boldsymbol{x}}_i, \hat{\boldsymbol{\kappa}}_i)) + \eta\mathcal{L}_{\text{APM}}, \quad (7)$$

where $\eta = 0.01$ is a hyperparameter to balance these two loss functions.

[1]The dimension of embeddings is determined by specific encoder architectures. When NFNet (Brock et al., 2021) and BERT (Devlin et al., 2019) are employed as the image and text encoders, their embedding dimensions are $d_I = 2078$ and $d_T = 768$, respectively, and the maximum size of the synthetic set $|\mathcal{S}|$ is 500.

## 3.3. Similarity Mining

In APM, we mainly focus on aligning the channel-level correspondence between the real and synthetic datasets, *i.e.*, the covariance matrices of multi-modal data. On the other hand, mining the correspondence between samples is also crucial for MDD, as pointed out by LoRS (Xu et al., 2024a). Specifically, LoRS uses a LoRA-like (Xu et al., 2024b) matrix, $Z = \omega I + LR^\top$, to record the similarities between samples and optimizes it during distillation.

In the evaluation stage, the similarity matrix is used to weight the binary cross-entropy loss function, aiding the training of multi-modal models. However, this method poses additional computational and space overhead for MDD. Different from LoRS, we directly use the teacher model to generate a similarity matrix of the synthetic pairs rather than training it, and adopt a knowledge distillation loss to train the model from scratch:

$$\mathcal{L}_{\text{KD}} = \sum_{i=1}^{|\mathcal{S}|} \sum_j P_{ij} \log\frac{P_{ij}}{Q_{ij}}, \quad (8)$$

$$P_i = \text{Softmax}(\tilde{Z}_i/\tau), \ Q_i = \text{Softmax}(Z_i/\tau),$$

where $\tilde{Z} = \mathcal{M}^t(\hat{X}, \hat{K})$ and $Z = \mathcal{M}^s(\hat{X}, \hat{K})$ are the similarity matrices learned by the teacher and student networks. We notice that the similarity matrix of APM is larger than that of LoRS. To address this issue, we can apply SVD on the similarity matrix and preserve eigenvectors corresponding to the top-$K$ singular values.

## 4. Experiments

In this section, we conduct extensive experiments to validate the effectiveness of our proposed method.

### 4.1. Experimental Setup

**Datasets and Metrics.** We first benchmark various MDD methods in two widely used image-text datasets: Flickr-30k (Plummer et al., 2015) and MS-COCO (Lin et al., 2014), where each image is paired with five human-annotated captions. We focus on the cross-modal retrieval task, which aims to retrieve the top-$K$ semantically relevant samples in the target modality conditioned on a query from the source modality. We use Recall at K (R@K) as the metric and consider two scenarios: image-to-text retrieval (TR@K) and text-to-image retrieval (IR@K). To ensure generalizability, we additionally use AutioCaps (Kim et al., 2019), an audio-text dataset, to evaluate the performance of MDD. See Appendix B.3 for more details.

**Networks.** We use a CLIP-style (Radford et al., 2021) network architecture as our distillation backbone, consisting of an image encoder, a text encoder, and two linear modal projectors. For the image encoder, we choose NFNet (Brock

*Table 2.* Results on MS-COCO dataset. We use NFNet+BERT as the distillation and evaluation networks. Full dataset performance: IR@1=11.1, IR@5=31.5, IR@10=44.7; TR@1=14.6, TR@5=37.6, TR@10=50.5. The best results are highlighted in bold.

| Pairs (Ratio) | Metric | Coreset Selection | | | | Dataset Distillation | | | | | |
|---|---|---|---|---|---|---|---|---|---|---|---|
| | | Rand | Herd | K-Cent | Forget | MTT-VL | TESLA | LoRS | CovMatch | RepBlend | APM |
| 100 (0.8‰) | IR@1 | 0.3 | 0.5 | 0.4 | 0.3 | 1.3±0.1 | 0.3±0.2 | 1.8±0.1 | 2.8±0.1 | 4.1±0.3 | **4.7±0.2** |
| | IR@5 | 1.3 | 1.4 | 1.4 | 1.5 | 5.4±0.3 | 1.0±0.4 | 7.1±0.2 | 10.5±0.2 | 13.9±0.8 | **16.2±0.2** |
| | IR@10 | 2.7 | 3.5 | 2.5 | 2.5 | 9.5±0.5 | 1.8±0.5 | 12.2±0.2 | 17.7±0.3 | 22.3±0.5 | **25.8±0.3** |
| | TR@1 | 0.8 | 0.8 | 1.4 | 0.7 | 2.5±0.3 | 2.0±0.2 | 3.3±0.2 | 3.8±0.1 | 5.2±0.5 | **6.2±0.4** |
| | TR@5 | 3.0 | 2.1 | 3.7 | 2.6 | 10.0±0.5 | 7.7±0.5 | 12.2±0.3 | 13.1±0.3 | 17.9±0.9 | **20.0±0.5** |
| | TR@10 | 5.0 | 4.9 | 5.5 | 4.8 | 15.7±0.4 | 13.5±0.3 | 19.6±0.3 | 21.1±0.2 | 28.0±0.3 | **31.1±0.5** |
| 200 (1.7‰) | IR@1 | 0.6 | 0.9 | 0.7 | 0.6 | 1.7±0.1 | 0.1±0.1 | 2.4±0.1 | 3.8±0.1 | **6.1±0.8** | 6.1±0.2 |
| | IR@5 | 2.3 | 2.4 | 2.1 | 2.8 | 6.5±0.4 | 0.2±0.1 | 9.3±0.2 | 13.4±0.1 | 19.3±0.7 | **19.6±0.2** |
| | IR@10 | 4.4 | 4.1 | 5.8 | 4.9 | 12.3±0.8 | 0.5±0.1 | 15.5±0.2 | 21.8±0.2 | 29.8±0.5 | **30.4±0.3** |
| | TR@1 | 1.0 | 1.0 | 1.2 | 1.1 | 3.3±0.2 | 0.7±0.2 | 4.3±0.1 | 5.3±0.2 | 6.9±0.6 | **7.7±0.5** |
| | TR@5 | 4.0 | 3.6 | 3.8 | 3.5 | 11.9±0.6 | 3.1±0.5 | 14.2±0.3 | 17.3±0.2 | 21.8±0.9 | **23.6±0.7** |
| | TR@10 | 7.2 | 7.7 | 7.5 | 7.0 | 19.4±1.2 | 5.3±0.8 | 22.6±0.2 | 27.0±0.2 | 32.3±0.7 | **35.3±0.9** |
| 500 (4.4‰) | IR@1 | 1.1 | 1.7 | 1.1 | 0.8 | 2.5±0.5 | 0.8±0.2 | 2.8±0.2 | 5.4±0.1 | 6.2±0.1 | **7.1±0.2** |
| | IR@5 | 5.0 | 5.3 | 6.3 | 5.8 | 8.9±0.7 | 3.6±0.6 | 9.9±0.5 | 18.0±0.1 | 19.9±0.3 | **21.8±0.3** |
| | IR@10 | 8.7 | 9.9 | 10.5 | 8.2 | 15.8±1.5 | 6.7±0.9 | 16.5±0.7 | 28.2±0.1 | 30.6±0.1 | **33.3±0.4** |
| | TR@1 | 1.9 | 1.9 | 2.5 | 2.1 | 5.0±0.4 | 1.7±0.4 | 5.3±0.3 | 8.1±0.3 | 7.0±0.2 | **8.0±0.4** |
| | TR@5 | 7.5 | 7.8 | 8.7 | 8.2 | 17.2±1.3 | 5.9±0.8 | 18.3±1.5 | 23.5±0.3 | 22.0±0.3 | **24.3±0.3** |
| | TR@10 | 12.5 | 13.7 | 14.3 | 13.0 | 26.0±1.9 | 10.2±1.0 | 27.9±1.4 | 34.6±0.6 | 32.9±0.6 | **37.1±0.4** |

et al., 2021), RegNet (Xu et al., 2023), ResNet-50 (He et al., 2016), and ViT (Dosovitskiy et al., 2021). For the text encoder, we use BERT (Devlin et al., 2019) and DistilBERT (Sanh et al., 2019). We directly optimize the synthetic images in the pixel space and update the embedding of the synthetic captions instead of the original text, as suggested by Wu et al. (2024). We use the officially pre-trained weights to initialize both the image and text encoders. During distillation and evaluation, both encoders are frozen, as suggested by Zhang et al. (2025).

**Baselines.** We benchmark APM with various MDD methods to demonstrate its effectiveness. Specifically, we consider two categories of methods: Coreset-based methods, including Random, Herding (Welling, 2009), K-Center, and Forgetting (Toneva et al., 2019), as well as the advanced distillation-based methods, including MTT-VL (Wu et al., 2024), LoRS (Xu et al., 2024a), CovMatch (Lee & Chung, 2025), and RepBlend (Zhang et al., 2025).

**Others.** Similar to LoRS, APM uses the similarity matrix to aid the training of models. For a fair comparison, we remove one synthetic pair to keep the total budget unchanged, *i.e.*, 100→99, 200→199, and 500→499. Moreover, we evaluate our methods five times and report the mean and standard deviation. See Appendix C for more details.

### 4.2. Main Results

Tables 2 and 3 report the performance on image-text datasets, from which we have the following observations:

- RepBlend performs best in the five trajectory matching baselines. It claims that matching the projector trajectories, rather than the full parameters, can improve the performance of MDD. This aligns with the ideas behind APM, where the modal projectors encode the essential features of a multimodal dataset.

- APM consistently outperforms existing methods in the IR@K metric. This advantage is not accidental: APM encourages embeddings to be more isotropic and better aligned across modalities, thereby reducing the semantic gap. As a result, it provides consistent improvements on the text-to-image side.

- APM has a superiority in large-scale datasets. We can observe that APM beats all baselines in the MS-COCO dataset, which has 123k images and is 4× larger than Flickr30k. Moreover, APM exhibits better scalability. When the budget increases from 200 to 500 pairs, RepBlend improves IR@1 marginally (6.1→6.2), while APM gains +1.0 (6.1→7.1).

We further benchmark these MDD methods on the audio-text dataset. Results are shown in Table 4. Following Zhang et al. (2025), we use EfficientAT (Schmid et al., 2023) and BERT as the audio and text encoders, respectively. We can observe that APM still outperforms LoRS and RepBlend, especially in audio retrieval tasks, where APM has 11.4 AR@1, while LoRS and RepBlend only achieve 7.1 and 9.7, respectively. The results demonstrate the superior generalization of APM across different multi-modal domains.

*Table 3.* Results on Flickr-30k dataset. We use NFNet+BERT as the distillation and evaluation networks. Full dataset performance: IR@1=21.3, IR@5=51.0, IR@10=63.6; TR@1=31.1, TR@5=61.7, TR@10=74.3. The best results are highlighted in bold.

| Pairs (Ratio) | Metric | Coreset Selection | | | | Dataset Distillation | | | | | |
| --- | --- | --- | --- | --- | --- | --- | --- | --- | --- | --- | --- |
| | | Rand | Herd | K-Cent | Forget | MTT-VL | TESLA | LoRS | CovMatch | RepBlend | APM |
| 100 (0.3%) | IR@1 | 1.0 | 0.7 | 0.7 | 0.7 | 4.7±0.2 | 0.5±0.2 | 8.3±0.2 | 10.1±0.2 | 11.5±0.4 | **12.8±0.4** |
| | IR@5 | 4.0 | 2.8 | 3.1 | 2.4 | 15.7±0.5 | 2.3±0.2 | 24.1±0.2 | 28.6±0.4 | 32.0±0.7 | **34.2±0.2** |
| | IR@10 | 6.5 | 5.3 | 6.1 | 5.6 | 24.6±1.0 | 4.7±0.4 | 35.1±0.3 | 40.9±0.6 | 44.5±0.6 | **47.1±0.3** |
| | TR@1 | 1.3 | 1.1 | 0.6 | 1.2 | 9.9±0.3 | 5.5±0.5 | 11.8±0.2 | 14.8±0.9 | 16.2±0.8 | **17.8±0.5** |
| | TR@5 | 5.9 | 4.7 | 5.0 | 4.2 | 28.3±0.5 | 19.5±0.9 | 35.8±0.6 | 38.0±0.4 | 41.7±0.9 | **43.0±1.2** |
| | TR@10 | 10.1 | 7.9 | 7.6 | 9.7 | 39.1±0.7 | 28.9±1.0 | 49.2±0.5 | 50.6±0.6 | 55.5±0.4 | **57.2±1.1** |
| 200 (0.7%) | IR@1 | 1.1 | 1.5 | 1.5 | 1.2 | 4.6±0.9 | 0.2±0.1 | 8.6±0.3 | 12.3±0.4 | 12.7±0.8 | **14.6±0.1** |
| | IR@5 | 4.8 | 5.5 | 5.4 | 3.1 | 16.0±1.6 | 1.3±0.2 | 25.3±0.3 | 33.6±0.3 | 34.7±0.6 | **38.5±0.2** |
| | IR@10 | 9.2 | 9.3 | 9.9 | 8.4 | 25.5±2.6 | 2.5±0.2 | 36.6±0.3 | 45.8±0.2 | 47.6±0.5 | **52.0±0.3** |
| | TR@1 | 2.1 | 2.3 | 2.2 | 1.5 | 10.2±0.8 | 2.8±0.5 | 14.5±0.5 | 17.4±0.5 | 18.6±0.7 | **18.9±1.2** |
| | TR@5 | 8.7 | 8.4 | 8.2 | 8.4 | 28.7±1.0 | 10.4±1.5 | 38.7±0.5 | 41.7±0.5 | 46.0±0.8 | **47.8±1.4** |
| | TR@10 | 13.2 | 14.4 | 13.5 | 10.2 | 41.9±1.9 | 17.4±1.6 | 53.4±0.5 | 55.8±0.5 | 60.0±0.6 | **62.2±1.1** |
| 500 (1.7%) | IR@1 | 2.4 | 3.0 | 3.5 | 1.8 | 6.6±0.3 | 1.1±0.2 | 10.0±0.2 | 14.7±0.3 | 17.0±0.6 | **17.5±0.3** |
| | IR@5 | 10.5 | 10.0 | 10.4 | 9.0 | 20.2±1.2 | 7.3±0.4 | 28.9±0.7 | 38.4±0.4 | 42.5±0.5 | **43.5±0.2** |
| | IR@10 | 17.4 | 17.0 | 17.3 | 15.9 | 30.0±2.1 | 12.6±0.5 | 41.6±0.6 | 51.4±0.3 | 55.9±0.6 | **56.8±0.3** |
| | TR@1 | 5.2 | 5.1 | 4.9 | 3.6 | 13.3±0.6 | 5.1±0.2 | 15.5±0.5 | 19.9±0.6 | **22.5±0.4** | 21.6±0.4 |
| | TR@5 | 18.3 | 16.4 | 16.4 | 12.3 | 32.8±1.8 | 15.3±0.5 | 39.8±0.6 | 46.7±0.9 | **53.2±0.3** | 52.7±0.2 |
| | TR@10 | 25.7 | 24.3 | 23.3 | 19.3 | 46.8±3.0 | 23.8±0.3 | 53.7±0.3 | 59.5±0.7 | **66.7±0.3** | 66.4±0.4 |

*Table 4.* Results on AudioCaps dataset. We use EfficientAT (mn20_as)+BERT as the distillation and evaluation networks. Full dataset performance: AR@1=17.6, AR@5=47.7, AR@10=63.8; TR@1=20.6, TR@5=49.6, TR@10=67.2. The best results are highlighted in bold.

| Pairs | Method | AR@1 | AR@5 | AR@10 | TR@1 | TR@5 | TR@10 |
| --- | --- | --- | --- | --- | --- | --- | --- |
| 100 | LoRS | 2.7±0.3 | 8.6±0.3 | 14.7±0.4 | 5.9±0.3 | 13.0±0.4 | 21.8±0.5 |
| | RepBlend | 4.1±0.2 | 14.2±0.3 | 23.7±0.4 | 8.9±0.1 | 24.3±0.2 | 34.7±0.3 |
| | APM | **8.3±0.3** | **28.6±0.3** | **42.1±0.4** | **11.3±0.6** | **33.4±0.6** | **46.7±0.8** |
| 200 | LoRS | 3.8±0.2 | 14.8±0.2 | 21.8±0.2 | 8.0±0.2 | 21.2±0.2 | 33.1±0.2 |
| | RepBlend | 6.8±0.2 | 20.6±0.2 | 31.4±0.3 | 9.7±0.2 | 29.1±0.4 | 41.2±0.4 |
| | APM | **10.1±0.1** | **32.5±0.3** | **47.3±0.2** | **11.7±0.7** | **35.6±0.8** | **51.1±1.1** |
| 500 | LoRS | 7.1±0.1 | 24.7±0.2 | 36.7±0.2 | 9.2±0.2 | 27.4±0.3 | 41.3±0.3 |
| | RepBlend | 9.7±0.1 | 32.2±0.3 | 46.8±0.2 | **13.8±0.3** | 38.6±0.3 | 54.1±0.4 |
| | APM | **11.4±0.1** | **35.8±0.4** | **51.3±0.3** | 13.6±0.7 | **39.3±0.7** | **54.8±0.5** |

*Table 5.* Ablation study on the loss function of APM under 100 pairs. "Random" means we randomly pick data for evaluation without training.

| Flickr | IR@1 | IR@5 | IR@10 | TR@1 | TR@5 | TR@10 |
| --- | --- | --- | --- | --- | --- | --- |
| Random | 3.4 | 11.5 | 18.5 | 4.1 | 12.8 | 21.3 |
| $+\mathcal{L}_{\text{NCE}}$ | 6.0 | 19.0 | 28.5 | 8.1 | 25.5 | 38.0 |
| $+\mathcal{L}_{\text{APM}}$ | **12.8** | **34.2** | **47.1** | **17.8** | **43.0** | **57.2** |

*Table 6.* Ablation study on the normalizations of the loss function.

| | IR@1 | IR@5 | IR@10 | TR@1 | TR@5 | TR@10 |
| --- | --- | --- | --- | --- | --- | --- |
| $\mathcal{L}_{\text{APM}}$ | 12.8 | 34.2 | 47.1 | 17.8 | 43.0 | 57.2 |
| w/o ES | 11.6 | 32.0 | 44.9 | 17.3 | 41.8 | 57.5 |
| w/o SE | 2.1 | 7.5 | 12.9 | 3.9 | 13.3 | 19.9 |
| w/o MI | 0.2 | 0.8 | 1.3 | 0.0 | 0.0 | 0.0 |

## 4.3. Ablation Studies

To further verify the effectiveness of each component in APM, we make ablation studies about the loss functions, normalizations, and hyperparameters.

**Loss Functions.** We first evaluate the role of loss functions, including $\mathcal{L}_{\text{NCE}}$ and $\mathcal{L}_{\text{APM}}$. The results are shown in Table 5. The randomly selected data gives a basic performance of MDD, validating the effectiveness of similarity mining. We further add the contrastive loss function $\mathcal{L}_{\text{NCE}}$ to optimize the synthetic data, which slightly improves the performance. Finally, we add $\mathcal{L}_{\text{APM}}$ in the distillation process, which significantly improves the IR@1 value from 6.0 to 12.8, demonstrating the superiority of APM.

**Normalizations.** In Section 3.2, we mention three issues

that affect the stability of the distillation process, including Embedding Shift (ES), Scale Explosion (SE), and Matrix Inversion (MI), and propose corresponding normalizations to address them. We further make an ablation study on the Flickr-30k dataset to validate the effectiveness of these normalizations. The results are shown in Table 6. We have the following observations: First, removing ES affects the performance of APM slightly. This is because synthetic data is initialized from real data and has similar average values. Second, removing SE significantly degenerates the retrieval performance because there is much more real data than synthetic data, resulting in different scales. Third, removing MI cannot obtain a meaningful synthetic dataset. Due to the

*Table 7.* Effect of hyperparameters in Flickr-30k.

| IR@1 $\backslash \eta$ $\alpha$ | 0.1 | 0.01 | 0.001 |
|---|---|---|---|
| 0.01 | 5.2 | 8.1 | 9.9 |
| 0.05 | 7.3 | **12.8** | 11.5 |
| 0.1 | 10.6 | 11.4 | 11.0 |

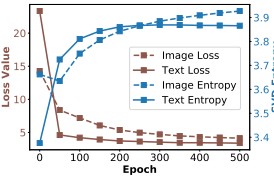

*Figure 3.* Loss and SVD entropy during distillation.

*Table 8.* Scalability experiments on Flickr-30k datasets. Results of MDW and EDGE are taken from Dang et al. (2025) and Zhao et al. (2025), while other results were implemented by ourselves.

| Pairs (Ratio) | Method | Flickr-30k | | | | | |
|---|---|---|---|---|---|---|---|
| | | IR@1 | IR@5 | IR@10 | TR@1 | TR@5 | TR@10 |
| 1000 (3.4%) | LoRS | 11.0 | 30.8 | 42.5 | 16.0 | 41.1 | 54.8 |
| | MDW | 12.5 | 32.2 | 45.8 | 19.2 | 49.1 | 63.0 |
| | EDGE | 9.9 | 28.2 | 40.5 | 14.5 | 38.3 | 51.7 |
| | RepBlend | 17.8 | 44.7 | 56.9 | 23.0 | **54.4** | **67.3** |
| | APM | **18.4** | **45.5** | **57.9** | **23.2** | 53.8 | 66.9 |

low-rank property of the synthetic dataset, directly solving for its inverse matrix will lead to numerical instability.

**Hyperparameters.** We next evaluate the influence of hyperparameters on the performance of APM. Specifically, we focus on two important hyperparameters: $\alpha$ in analytic parameters and $\eta$ in loss functions. We can observe from Table 7 that the best result is obtained with $\eta = 0.01$ and $\alpha = 0.05$. Generally, the hyperparameter $\alpha$ controls the frequency of the covariance matrix (Bo et al., 2026). A smaller value of $\alpha$ introduces more high-frequency noise, while a large value of $\alpha$ makes the images blurred. On the other hand, a larger value of $\eta$ may enforce the synthetic dataset to overfit the real analytic parameters, and a smaller value cannot narrow the distribution gap between the real and synthetic datasets.

### 4.4. In-depth Analysis

**Data Entropy.** The goal of DD is to reduce the redundancy in the real datasets. To verify whether APM can achieve this objective, we analyze the entropy of the image and text embeddings in the synthetic dataset. Specifically, we use the SVD entropy, which is defined as $\mathcal{H} = -\sum_i \sigma_i \log \sigma_i$, where $\sigma_i = \frac{s_i}{\sum_i s_i}$ and $s_i$ denote the $i$-th singular value of the data embeddings. Intuitively, data embeddings with smaller SVD entropy have more redundancy as their information is dominated by a few principal singular values, and vice versa. Based on this property, we draw the trends of loss and SVD entropy of the image and text embeddings in Figure 3. We can observe that as the loss function decreases, the SVD entropy of the data embeddings gradually increases, implying that APM can effectively reduce data redundancy and improve data diversity.

*Table 9.* Cross-architecture performance of various MDD methods in the Flickr-30k with 500 pairs. The synthetic dataset is distilled on NFNet+BERT and evaluated by other networks.

| Evaluation | Method | IR@1 | IR@5 | IR@10 | TR@1 | TR@5 | TR@10 |
|---|---|---|---|---|---|---|---|
| ResNet +BERT | TESLA | 3.0±0.2 | 10.8±0.5 | 17.0±0.8 | 6.0±0.9 | 18.8±0.7 | 27.7±1.2 |
| | LoRS | 3.3±0.2 | 12.7±0.3 | 20.4±0.2 | 6.8±0.2 | 19.6±1.3 | 31.1±0.3 |
| | RepBlend | 4.2±0.2 | 14.1±0.2 | 23.6±0.6 | 8.4±0.2 | 23.1±0.8 | 35.0±1.3 |
| | APM | **6.9±0.2** | **21.2±0.3** | **31.2±0.4** | **8.7±0.7** | **24.5±0.5** | **35.9±1.2** |
| RegNet +BERT | TESLA | 3.2±0.8 | 11.1±1.8 | 17.5±1.3 | 5.8±0.1 | 18.6±0.6 | 28.1±1.0 |
| | LoRS | 3.5±0.1 | 12.6±0.3 | 21.1±0.4 | 6.8±0.3 | 20.8±0.3 | 30.2±0.3 |
| | RepBlend | 3.9±0.2 | 13.9±0.3 | 24.0±0.6 | **7.9±0.3** | **24.2±0.3** | **36.2±1.1** |
| | APM | **5.4±0.1** | **16.7±0.4** | **25.3±0.5** | 7.9±0.5 | 22.2±0.6 | 32.1±0.6 |

**Scalability.** In addition to the efficacy and efficiency, we also emphasize the scalability of the method, as lossless performance is only possible on relatively large-scale synthetic datasets. For example, in the setting of Flickr-30k with 500 pairs, the results of APM are 17.5 in IR@1 and 21.6 in TR@1, which are still far behind the performance on the full dataset (21.3 in IR@1 and 31.1 in TR@1). To evaluate the scalability of APM, we increase the maximum budget from 500 pairs to 1,000 pairs. The results are listed in Table 8. It can be observed that APM achieves the best results in 4 of 6 metrics, while only slightly outperformed by RepBlend in TR@5 and TR@10, demonstrating its scalability.

### 4.5. Generalization Tasks

**Cross-Architecture Generalization.** Finally, we evaluate the cross-architecture generalization of different MDD methods. Following previous work (Zhang et al., 2025), we use NF-ResNet-50 and NF-RegNet as the image encoders, respectively, and BERT as the text encoder. The results are shown in Table 9, from which we can find that APM exhibits the strongest generalization ability across architectures. First, when evaluated on ResNet+BERT, APM achieves the best performance on all metrics, *e.g.*, 6.9 IR@1 and 8.7 TR@1, surpassing RepBlend by +2.7 and +0.3, respectively. Second, on RegNet+BERT, APM consistently outperforms the baselines, reaching 5.4 IR@1 and 7.9 TR@1, while the second-best method only achieves 3.9 and 6.8. This demonstrates that APM not only learns compact and effective synthetic datasets but also transfers well to unseen architectures. The results validate our claim that APM preserves the essential modality alignment in a way that is independent of specific backbone choices, highlighting its scalability and robustness for real-world deployment.

**Zero-shot Classification.** To verify whether the synthetic dataset can be used in downstream tasks beyond retrieval. We make a zero-shot image classification task to benchmark the performance between real and synthetic datasets. Specifically, we use three datasets, CIFAR-10, CIFAR-100, and ImageNet-1k. The results are shown in Table 10. We can see that the synthetic datasets have similar zero-shot classification performance to the real dataset.

*Table 10.* Results of zero-shot image classification.

| Dataset | CIFAR-10 | | CIFAR-100 | | ImageNet-1k | |
| --- | --- | --- | --- | --- | --- | --- |
| | Top-1 (%) | Top-5 (%) | Top-1 (%) | Top-5 (%) | Top-1 (%) | Top-5 (%) |
| Full | 58.77 | 92.07 | 16.34 | 38.27 | 7.62 | 19.54 |
| 99 Pairs | 52.44 | 87.54 | 13.53 | 32.19 | 4.28 | 12.50 |
| 199 Pairs | 53.73 | 85.96 | 13.70 | 34.76 | 4.35 | 12.69 |
| 499 Pairs | **55.03** | **90.28** | **14.45** | **36.14** | **5.06** | **14.77** |

## 4.6. Visualization

**Modality Distribution.** A recent work (Zhang et al., 2025) highlights that MDD methods suffer from the modality collapse issue, where intra-modality embeddings are overly concentrated, while cross-modality embeddings are not well aligned. To verify whether APM can address this issue, we project the image and text embeddings on a spherical surface, as shown in Figure 4. We can observe that the image and text embeddings are well-matched in the shared space, indicating that APM can preserve the data correspondence across modalities.

**Synthetic Image-text Pairs.** We compare the real dataset with the synthetic dataset learned by APM in Figure 5. To be more intuitive, the synthetic data pairs are initialized by the real data in Flickr-30k. It can be observed that the caption learned by APM contains more detailed descriptions, such as the clothing of people. Moreover, the images also have high-frequency artifacts. We speculate that these textures will increase the diversity of data.

**Synthetic Log-spectrogram.** Figure 6 illustrates the original and distilled log-mel-spectrograms in the AudioCaps dataset. We can see that the distilled log-mel-spectrogram has higher energy than the original one, indicating that it compresses the knowledge of real audios.

## 5. Related Work

**Dataset Distillation.** The concept of dataset distillation (DD) was first introduced by Wang et al. (2018), with the goal of condensing a large dataset into a compact set of synthetic samples while maintaining comparable performance. Existing methods can be broadly categorized into three groups: gradient matching (Zhao et al., 2021; Kim et al., 2022), which aligns gradients computed on real and synthetic data; trajectory matching (Cazenavette et al., 2022; Guo et al., 2024), which supervises the student's optimization trajectory using expert trajectories trained on real data; and statistical matching (Zhao & Bilen, 2023), which aligns higher-order statistics such as feature distributions or batch normalization statistics (Yin et al., 2023; Shao et al., 2024). Moreover, UniDD (Bo et al., 2026) provides a unified spectral filtering view of DD, under which our proposed APM can also be interpreted as a high-pass filter. DD has also been applied across diverse domains, including

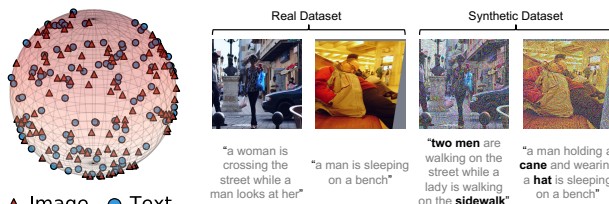

*Figure 4.* Distribution of synthetic dataset (Flickr).

*Figure 5.* The real and synthetic data pairs of APM. We highlight some fine-grained descriptions in the synthetic captions.

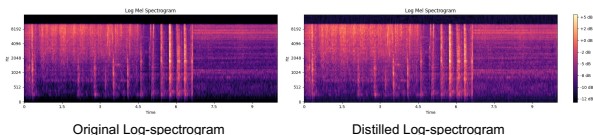

*Figure 6.* Visualizations of the original and distilled audio data.

images (Zhao et al., 2021; Yin et al., 2023), time series (Liu et al., 2024b; Ding et al., 2024), and graphs (Jin et al., 2022; Liu et al., 2024a).

**Multi-modal Dataset Distillation.** Compared with single-modal distillation, the multi-modal setting introduces additional challenges, as it requires preserving both intra-modal semantics and cross-modal alignment. Recent studies have extended trajectory matching (TM) to the multi-modal domain. For instance, MTT-VL (Wu et al., 2024) proposes bi-trajectory matching to align the paired image-text data. LoRS (Xu et al., 2024a) further introduces the concept of similarity mining, improving the performance of MDD by a large margin. More recently, RepBlend (Zhang et al., 2025) identifies the issue of modality collapse in MDD and proposes representation blending to preserve cross-modal consistency. MDW (Dang et al., 2025) further investigates the robustness of MDD under noisy environments. EDGE (Zhao et al., 2025) improves the efficiency and scalability of MDD by leveraging the prior knowledge of generative models. In addition to visual-language datasets, there are studies in other fields, such as audio and video (Li et al., 2025; Kushwaha et al., 2024).

## 6. Conclusion

In this paper, we introduce APM, a framework that improves the efficiency and scalability of multi-modal dataset distillation. APM uses the analytic parameters of linear modal projectors to replace the inner-loop optimization in trajectory matching, enabling efficient alignment of real and synthetic datasets. Extensive experiments on Flickr30k and MS-COCO demonstrate that APM not only reduces both storage and computational overhead but also maintains superior performance. A promising future direction is to extend APM to other modalities, such as audio-visual datasets.

## Acknowledgment

This research is supported by the Ministry of Education, Singapore, under the Academic Research Fund Tier 1 (FY2026).

## Impact Statement

This paper presents work whose goal is to advance the field of Machine Learning. In particular, it aims to accelerate the training of deep neural networks and reduce storage overhead. There are many potential societal consequences of our work, none of which we feel must be specifically highlighted here.

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

# A. Derivation

## A.1. Analytic Solution of Linear Projector under InfoNCE Loss

In multi-modal contrastive learning, we typically adopt the InfoNCE loss to optimize models:

$$\mathcal{L}_{\text{NCE}} = -\frac{1}{|\mathcal{D}|} \sum_{i=1}^{|\mathcal{D}|} \log \frac{\exp(u_i v_i^\top)}{\sum_{j=1}^{|\mathcal{D}|} \exp(u_i v_j^\top)}. \tag{9}$$

However, the nonlinear softmax activation function used in InfoNCE presents challenges for calculating the analytic projector parameters.

We begin by introducing an existing result about the analytic solution of a linear layer with the softmax function in the multi-class classification task:

**Lemma A.1.** *The probability that a sample $x$ belongs to a certain class $i$ is defined as:*

$$p(i|x) = \frac{\exp(x w_i^\top + b_i)}{\sum_{i=1}^{k} \exp(x w_i^\top + b_i)}, \tag{10}$$

*where $w_i$ and $b_i$ denote the weight and bias of the $i$-th class, respectively. The analytic solutions of $w_i$ and $b_i$ are defined as:*

$$w_i = \mu_i \Sigma^{-1}, \quad b_i = \ln p_i - \frac{1}{2} \mu_i \Sigma^{-1} \mu_i^\top, \tag{11}$$

*where $p_i$ is the ratio of the $i$-th class, $\mu_i$ is the mean value of the data embedding in the $i$-th class. Notably, $\Sigma = \hat{\Sigma} + \hat{\mu}^\top \hat{\mu} + \sum_i p_i \mu_i^\top \mu_i$ is an estimation of the ground-truth data covariance, where $\hat{\mu} = \frac{1}{N} \sum_{j=1}^{N} x_j$ represents the mean of all samples, and $\hat{\Sigma} = \frac{1}{N} \sum_{j=1}^{N} (x_i - \hat{\mu})^\top (x_i - \hat{\mu})$ denotes the covariance of all sample.*

*Proof.* See Equations 12-15 in Su (2021). □

Notably, the InfoNCE loss can also be viewed as a multi-class classification task, where each pair is a class. In this case, $u_i$ can be seen as the sample and $v_i^\top$ denotes the weight in the $i$-th class. Therefore, we have the following theoretical results (Proposition 3.1):

**Proposition A.2.** *For the linear projectors $U = H_I W_I$ and $V = H_T W_T$, $\mathcal{L}_{NCE}$ has analytical solutions with respect to $W_I$ and $W_T$, defined as:*

$$W_I^* = \frac{|\mathcal{D}|}{2\tau} \underbrace{\left(H_I^\top H_I\right)^{-1} H_I^\top}_{\textit{Image}} \underbrace{V \left(V^\top V\right)^{-1}}_{\textit{Text}}, \tag{12}$$

$$W_T^* = \frac{|\mathcal{D}|}{2\tau} \underbrace{\left(H_T^\top H_T\right)^{-1} H_T^\top}_{\textit{Text}} \underbrace{U \left(U^\top U\right)^{-1}}_{\textit{Image}}. \tag{13}$$

*Proof.* According to Lemma A.1, we can directly obtain the analytic solutions of $U$ and $V$:

$$u_i = \frac{1}{\tau} v_i \Sigma_V^{-1}, \quad v_i = \frac{1}{\tau} u_i \Sigma_U^{-1}, \tag{14}$$

where $\Sigma_V$ and $\Sigma_U$ denote the ground-truth covariance of each modality, where

$$\Sigma_V = \hat{\Sigma}_V + \hat{\mu}_V^\top \hat{\mu}_V + \sum_i p_i v_i^\top v_i \tag{15}$$

$$= \frac{2}{|D|} \sum_i v_i^\top v_i = \frac{2}{|D|} V^\top V. \tag{16}$$

Therefore, $\Sigma_V^{-1} = \frac{|D|}{2}(U^\top U)^{-1}$ and $\Sigma_U^{-1} = \frac{|D|}{2}(V^\top V)^{-1}$. We then transform this equation into matrix form by stacking a series of vectors, and obtain:

$$U = \frac{|D|}{2\tau}V(V^\top V)^{-1}, \quad V = \frac{|D|}{2\tau}U(U^\top U)^{-1}. \tag{17}$$

Based on the above results, the analytic parameters of linear projectors can be calculated via the pseudo-inverse of $H_I$ and $H_T$:

$$W_I^* = \frac{|D|}{2\tau}(H_I^\top H_I)^{-1}H_I^\top V(V^\top V)^{-1}, \tag{18}$$

$$W_T^* = \frac{|D|}{2\tau}(H_T^\top H_T)^{-1}H_T^\top U(U^\top U)^{-1}. \tag{19}$$

□

## A.2. Analytic Solution of Non-linear Projector

We further generalize our theoretical results from the linear projectors to non-linear projectors $U = \sigma(H_I W_I)$ and $V = \sigma(H_T W_T)$, and have the the following conclusion (Proposition 3.2):

**Proposition A.3.** *For non-linear projectors $U = \sigma(H_I W_I)$ and $V = \sigma(H_T W_T)$, the analytic solutions becomes:*

$$W_I^* = (H_I^\top H_I)^{-1}H_I^\top \sigma^{-1}(\frac{|\mathcal{D}|}{2\tau}V(V^\top V)^{-1}), \tag{20}$$

$$W_T^* = (H_T^\top H_T)^{-1}H_T^\top \sigma^{-1}(\frac{|\mathcal{D}|}{2\tau}U(U^\top U)^{-1}), \tag{21}$$

*where $\sigma^{-1}(\cdot)$ is the inverse function of the activation function $\sigma(\cdot)$.*

*Proof.* Equation 17 gives the analytic solutions of modal representations after the projectors, *i.e.*, $U$ and $V$. We can use the inverse function of the activation to transform the nonlinear projectors to linear projectors:

$$\sigma^{-1}(U) = H_I W_I, \quad \sigma^{-1}(V) = H_T W_T. \tag{22}$$

Then we can use the pseudo-inverse of $H_I$ and $H_T$ to calculate the analytic projector parameters:

$$W_I^* = (H_I^\top H_I)^{-1}H_I^\top \sigma^{-1}(\frac{|\mathcal{D}|}{2\tau}V(V^\top V)^{-1}), \tag{23}$$

$$W_T^* = (H_T^\top H_T)^{-1}H_T^\top \sigma^{-1}(\frac{|\mathcal{D}|}{2\tau}U(U^\top U)^{-1}), \tag{24}$$

□

We list some commonly used activation functions and their inverses below.

*Table 11.* Activation Functions $\sigma(x)$ and their inverses $\sigma^{-1}(y)$

| **Activation** | $\sigma(x)$ | $\sigma^{-1}(y)$ |
|---|---|---|
| Sigmoid | $\sigma(x) = \dfrac{1}{1 + e^{-x}}$ | $\sigma^{-1}(y) = \ln\left(\dfrac{y}{1-y}\right)$ |
| Tanh | $\sigma(x) = \tanh(x) = \dfrac{e^x - e^{-x}}{e^x + e^{-x}}$ | $\sigma^{-1}(y) = \mathrm{arctanh}(y) = \dfrac{1}{2}\ln\left(\dfrac{1+y}{1-y}\right)$ |
| LeakyReLU | $\sigma(x) = \begin{cases} x, & x \geq 0 \\ \alpha x, & x < 0 \end{cases}$ | $\sigma^{-1}(y) = \begin{cases} y, & y \geq 0 \\ \frac{y}{\alpha}, & y < 0 \end{cases}$ |

# B. Additional Experiments

## B.1. Intra-covariance and Inter-covariance

We make an experiment to verify the effectiveness of simultaneously distilling the intra- and inter-covariance information of real datasets. As mentioned in Section 3.1, the analytic parameters $(H_I^\top H_I)^{-1} H_I^\top V (V^\top V)^{-1}$ contain both intra-covariance and inter-covariance, while CovMatch (Lee & Chung, 2025), $U^\top V$, only explores the inter-covariance matching.

The comparison between the APM and CovMatch is shown in Table 12. We have the following observations: First, the synthetic data learned by APM has large entropy, indicating that it can encode more information about the real data. Second, APM outperforms CovMatch by a large margin, verifying the effectiveness of APM and supporting our claims.

*Table 12.* Comparison between isotropy and anisotropy distributions.

| Method | Equation | Image Entropy | Text Entropy | IR@1 | TR@1 |
|---|---|---|---|---|---|
| APM | $(H_I^\top H_I)^{-1} H_I^\top V (V^\top V)^{-1}$ | 3.93 | 3.87 | 12.8 | 17.8 |
| CovMatch | $U^\top V$ | 3.87 | 3.79 | 10.1 | 14.8 |

## B.2. Comparison with EDGE

EDGE leverages generative models to address the semantic correlation and diversity issues of existing MDD methods. Notably, we have cited EDGE in the original paper.

EDGE mainly focuses on the large budget setting (500 / 1000 pairs). We report the performance of EDGE and APM under the same settings in the revision. Below is a quick comparison.

*Table 13.* Flickr Retrieval Results

| Flickr | IR@1 | IR@5 | IR@10 | TR@1 | TR@5 | TR@10 |
|---|---|---|---|---|---|---|
| EDGE-500 | 6.7 | 21.0 | 30.5 | 13.3 | 35.6 | 47.5 |
| APM-500 | **17.5** | **43.5** | **56.8** | **21.6** | **52.7** | **66.4** |
| EDGE-1000 | 9.9 | 28.2 | 40.5 | 14.5 | 38.3 | 51.7 |
| APM-1000 | **18.4** | **45.5** | **57.9** | **23.2** | **53.8** | **66.9** |

*Table 14.* MS-COCO Retrieval Results

| MS-COCO | IR@1 | IR@5 | IR@10 | TR@1 | TR@5 | TR@10 |
|---|---|---|---|---|---|---|
| EDGE-500 | 1.8 | 6.5 | 11.2 | 2.9 | 9.5 | 15.7 |
| APM-500 | **7.1** | **21.8** | **33.3** | **8.0** | **24.3** | **37.1** |

## B.3. Audio-text Dataset Distillation

We make an additional experiment on the audio-text retrieval task to verify the generalization of the proposed method. Following Zhang et al. (2025), we choose the AudioCaps (Kim et al., 2019) dataset, consisting of 49,838 training audios, 495 validation audios, and 975 test audios. We use EfficientAT (mn20_as) (Schmid et al., 2023) as the audio encoder and BERT as the text encoder. Since RepBlend (Zhang et al., 2025) does not introduce its implementation details, we describe our reproduction process below.

**Data Preparation.** The AudioCaps dataset contains files in WAV format. We use the *AugmentMelSTFT* function from EfficientAT to preprocess the audio. We sample the audios in mono at a sampling rate of 32 kHz and then calculate their log-mel-spectrogram in a 25-ms window with a step size of 10 ms. After processing, each audio has a feature map with shape [1, 128, 1000], as shown in Figure 6.

**Distillation.** Instead of distilling the raw audios, we directly synthesize the log-mel-spectrogram to match the input of EfficientAT. The spectrogram has a shape of [1, 128, 1000], which can be seen as an image with channel=1, width=128,

*Table 15.* Hyperparameters used in the distillation stage.

| Dataset | Flickr | | | COCO | | | AudioCaps | | |
|---|---|---|---|---|---|---|---|---|---|
| Pairs | 100 | 200 | 500 | 100 | 200 | 500 | 100 | 200 | 500 |
| Epoch | 400 | 400 | 400 | 400 | 400 | 400 | 400 | 400 | 400 |
| Optimizer | Adam | Adam | Adam | Adam | Adam | Adam | Adam | Adam | Adam |
| LR | 0.1 | 0.1 | 0.1 | 0.1 | 0.1 | 0.1 | 0.01 | 0.01 | 0.01 |
| Betas | (0.6, 0.9) | (0.6, 0.9) | (0.6, 0.9) | (0.6, 0.9) | (0.6, 0.9) | (0.6, 0.9) | (0.6, 0.9) | (0.6, 0.9) | (0.6, 0.9) |
| $\alpha$ | 0.05 | 0.05 | 0.05 | 0.05 | 0.05 | 0.05 | 0.1 | 0.1 | 0.1 |
| $\eta$ | 0.01 | 0.01 | 0.01 | 0.01 | 0.01 | 0.01 | 0.01 | 0.01 | 0.01 |
| Projector Dim. | 256 | 256 | 256 | 256 | 256 | 256 | 256 | 256 | 256 |

*Table 16.* Hyperparameters used in the evaluation stage.

| Dataset | Flickr | | | COCO | | | AudioCaps | | |
|---|---|---|---|---|---|---|---|---|---|
| Pairs | 100 | 200 | 500 | 100 | 200 | 500 | 100 | 200 | 500 |
| Epoch | 100 | 100 | 100 | 100 | 100 | 100 | 100 | 100 | 100 |
| Optimizer | SGD | SGD | SGD | SGD | SGD | SGD | SGD | SGD | SGD |
| LR | 0.1 | 0.1 | 0.1 | 0.1 | 0.1 | 0.1 | 0.1 | 0.1 | 0.1 |
| Momentum | 0.9 | 0.9 | 0.9 | 0.9 | 0.9 | 0.9 | 0.9 | 0.9 | 0.9 |
| Weight Decay | 0.0005 | 0.0005 | 0.0005 | 0.0005 | 0.0005 | 0.0005 | 0.0005 | 0.0005 | 0.0005 |
| Scheduler | StepLR | StepLR | StepLR | StepLR | StepLR | StepLR | StepLR | StepLR | StepLR |
| Projector Dim. | 256 | 256 | 512 | 256 | 256 | 512 | 256 | 256 | 256 |
| KD Temperature ($\tau$) | 5 | 5 | 10 | 5 | 5 | 10 | 5 | 5 | 5 |

and height=1000. Therefore, the code of image-text distillation can be directly transferred to the audio-text datasets. See Table 15 for the hyperparameters.

**Evaluation.** In the test set of AudioCaps, each audio corresponds to 5 captions, which improves its retrieval performance. We train the multi-modal network from scratch and evaluate it on the test set. We repeat the experiments five times and report the average performance and standard deviation.

**Results.** The results are shown in Table 4. We can observe that APM outperforms LoRS and RepBlend, especially in audio retrieval tasks, where it shows a significant performance improvement. Figure 6 illustrates the original and distilled log-mel-spectrograms. We can see that the distilled log-mel-spectrogram has more energy than the original one, indicating that it compresses the knowledge of other audios.

## C. Experimental Details

**Preprocessing.** The derivation of the analytic parameters of modal projectors is based on the one-to-one correspondence between images and text. However, in Flickr-30k and MS-COCO, the ratio of the number of images to captions is 1:5, which makes it impossible to directly use APM. To address this issue, we uniformly divide the captions into five datasets and ensure that each image has a corresponding caption. During distillation, we cyclically select one sub-dataset to participate in the calculation of the real analytic parameter, thereby preventing the overfitting of the synthetic dataset.

**Hyperparameters.** To improve the reproducibility of our work, we provide the hyperparameters used in both distillation and evaluation stages in Tables 15 and 16.

**Algorithms** Algorithm 1 illustrates the distillation process of APM. Algorithm 2 shows the Pytorch-style code of APM.

---

**Algorithm 1** Analytic Parameter Matching (APM)

---

**input** Distillation network $\mathcal{M} = \{f_{\mathrm{E}}, f_{\mathrm{P}}, g_{\mathrm{E}}, g_{\mathrm{P}}\}$, real dataset $\mathcal{D} = (X, K)$, number of iteration $\mathcal{I}$.
**output** Synthetic dataset $\mathcal{S} = (\hat{X}, \hat{K})$
1: Feed $\mathcal{D}$ into $\mathcal{M}$, where $H_I = f_{\mathrm{E}}(X), H_T = g_{\mathrm{E}}(K), U = g_{\mathrm{E}}(H_I), V = g_{\mathrm{P}}(H_T)$
2: Calculate $\Sigma_{II}, \Sigma_{UU}, \Sigma_{IV}, \Sigma_{TU}, \Sigma_{TT}$, and $\Sigma_{VV}$
3: Calculate $W_I^* = \Sigma_{II}^{-1} \Sigma_{IV} \Sigma_{VV}^{-1}$ and $W_T^* = \Sigma_{TT}^{-1} \Sigma_{TU} \Sigma_{UU}^{-1}$
4: **for** iteration $i = 1, \cdots, \mathcal{I}$ **do**
5:     Feed $\mathcal{S}$ into $\mathcal{M}$, where $\hat{H}_I = f_{\mathrm{E}}(\hat{X}), \hat{H}_T = g_{\mathrm{E}}(\hat{K}), \hat{U} = g_{\mathrm{E}}(\hat{H}_I), \hat{V} = g_{\mathrm{P}}(\hat{H}_T)$
6:     Calculate $\hat{\Sigma}_{II}, \hat{\Sigma}_{UU}, \hat{\Sigma}_{IV}, \hat{\Sigma}_{TU}, \hat{\Sigma}_{TT}$, and $\hat{\Sigma}_{VV}$
7:     Calculate $\hat{W}_I^* = \hat{\Sigma}_{II}^{-1} \hat{\Sigma}_{IV} \hat{\Sigma}_{VV}^{-1}$ and $\hat{W}_T^* = \hat{\Sigma}_{TT}^{-1} \hat{\Sigma}_{TU} \hat{\Sigma}_{UU}^{-1}$
8:     Minimize the discrepancy between analytic parameters based on Equation 6
9: **end for**

---

**Algorithm 2** PyTorch code of APM

---

```
1   def Conv(img_embed, txt_embed, img_proj, txt_proj, alpha=0.1):
2       device = img_embed.device
3       N = img_embed.shape[0]
4
5       h_I = img_embed - img_embed.mean(0, keepdim=True)
6       h_T = txt_embed - txt_embed.mean(0, keepdim=True)
7       h_U = img_proj - img_proj.mean(0, keepdim=True)
8       h_V = txt_proj - txt_proj.mean(0, keepdim=True)
9
10      sigma_II = (h_I.T @ h_I) / N + alpha * torch.eye(h_I.shape[1], device=device)
11      sigma_IV = (h_I.T @ h_V) / N
12      sigma_VV = (h_V.T @ h_V) / N + alpha * torch.eye(h_V.shape[1], device=device)
13
14      tmp = torch.linalg.solve(sigma_II, sigma_IV)
15      w_I = torch.linalg.solve(sigma_VV, tmp, left=False)
16
17      sigma_TT = (h_T.T @ h_T) / N + alpha * torch.eye(h_T.shape[1], device=device)
18      sigma_TU = (h_T.T @ h_U) / N
19      sigma_UU = (h_U.T @ h_U) / N + alpha * torch.eye(h_U.shape[1], device=device)
20
21      tmp2 = torch.linalg.solve(sigma_TT, sigma_TU)
22      w_T = torch.linalg.solve(sigma_UU, tmp2, left=False)
23
24      return w_I, w_T
```

---

