# OpenReview forum: "Efficient Multi-modal Dataset Distillation via Analytic Parameter Matching"
_ICML.cc/2026/Conference — ICML 2026 regular_

### Official Review · Reviewer_5gbg · 2026-03-04

**Soundness:** 3
**Presentation:** 3
**Significance:** 2
**Originality:** 3
**Overall Recommendation:** 4
**Confidence:** 2

**Summary:**

The paper addresses the efficiency and scalability bottlenecks in Multi-modal Dataset Distillation (MDD). Traditional methods rely on Trajectory Matching (TM), which requires storing extensive teacher model checkpoints and performing computationally expensive bi-level optimization. The authors propose Analytic Parameter Matching (APM), a framework that replaces the inner-loop optimization by theoretically deriving the analytic (closed-form) solutions for modal projectors. By minimizing the discrepancy between the analytic parameters of the teacher and student models, APM achieves significant speedup and memory reduction.

**Compliance With Llm Reviewing Policy:**

Affirmed.

**Final Justification:**

I maintain my positive score.

**Key Questions For Authors:**

See Strengths And Weaknesses.

**Limitations:**

No, the authors should add that.

**Strengths And Weaknesses:**

Pros:
1. The shift from iterative gradient-based matching to analytic parameter alignment is a novel perspective in the distillation community.
2. The method is validated across multiple modalities (Image-Text and Audio-Text) and demonstrates promising generalization and performance.

Cons:
1. I noticed that alpha takes different values on different datasets and does not follow the optimal settings of Flickr-30k. Therefore, I have concerns about the generalizability of its method. Is it necessary to conduct grid experiments on different datasets to select the optimal parameters?
2. In practice, the proposed method relies on matrix factorization. If the feature dimensions are large, does its efficiency advantage still hold?
3. The authors primarily focused on cross-modal retrieval as a validation task, and related work should be reviewed as much as possible.

---

> ### Author Rebuttal · Authors · 2026-03-31
>
> We sincerely appreciate your thoughtful feedback and insightful questions.
>
> &nbsp;
>
> > **W1: Hyperparameters of APM**
>
> A1: The hyper-parameter $\alpha$ controls the data entropy of the synthetic pairs, which is essential for the performance of APM. As the data entropy varies across different datasets, it is necessary to try different values of $\alpha$.
>
> In the experiments, we only try three different values of $\alpha$, including 0.01, 0.05, and 0.1. Although the grid search slightly increases the complexity of the experiment, it is acceptable for the sake of performance improvement.
>
> On the other hand, we find that **the value of $\alpha$ is agnostic to the number of synthetic pairs**. For example, we choose $\alpha=0.05$ for 100, 200, and 500 pairs in the Flickr and COCO datasets. This indicates that APM has a generalization over synthetic pairs. We can search for the optimal $\alpha$ under smaller synthetic datasets and generalize it to larger datasets to reduce computational costs.
>
> &nbsp;
>
> > **W2: Efficiency of APM with respect to feature dimensions**
>
> A2: We truly understand your concerns about the matrix inversion. Empirically, we find that **the efficiency of APM is bounded by the number of synthetic pairs rather than the dimension of features**.
>
> ``[Experiment]`` To verify this, we evaluate the time and space overhead of APM on the Flickr dataset. Specifically, we vary the dimension of projectors from 256 to 2,048, and the number of synthetic pairs increases from 99 to 299. The results are shown below
>
> |Time (s/iter)|dim=256|dim=512|dim=1024|dim=2048|
> |-|-|-|-|-|
> |APM (99 pairs)|0.097|0.099|0.102|0.111|
>
> |Space (MB)|dim=256|dim=512|dim=1024|dim=2048|
> |-|-|-|-|-|
> |APM (99 pairs)|10710|10720|10790|10918|
>
> |Time (s/iter)|99 pairs|199 pairs|299 pairs|
> |-|-|-|-|
> |APM (dim=256)|0.097|0.175|**0.256**|
>
> |Space (MB)|99 pairs|199 pairs|299 pairs|
> |-|-|-|-|
> |APM (dim=256)|10710|19120|**27968**|
>
> ``[Reason]`` We can find that enlarging the dimension of modal projectors only slightly increases the time and space overhead of APM. The reasons are two-fold
>
> - The complexity of matrix inversion, e.g., $(V^{\top}V)^{-1}$, depends on the rank of $V^{\top}V$. As we mentioned in Lines 166-168, the largest number of synthetic pairs is 500, which is relatively smaller than the dimension of features. As a result, the rank of $V^{\top}V$ is bounded by the number of synthetic pairs.
>
> - For the real dataset with a large number of samples, we will calculate its matrix inversion **offline**, which will not affect the efficiency of APM during distillation.
>
>
> &nbsp;
>
> > **W3: Related works about cross-modal retrieval**
>
> A3: In the revision, we will strengthen the related work part by clarifying the following development line:
>
> - CLIP [1] established the large-scale dual-encoder contrastive learning paradigm, enabling efficient cross-modal retrieval through a shared image-text embedding space.
> - ALBEF [2] and BLIP [3] improved upon this framework by introducing stronger cross-modal alignment and fusion mechanisms, which enhanced fine-grained semantic matching between images and text.
> - BLIP-2 [4] and SigLIP [5] further extended this line toward more general and scalable vision-language modeling, improving the quality of retrieval-oriented representations and connecting retrieval with broader VLM capabilities.
>
> This revision will help better position our work within the evolution of cross-modal retrieval research.
>
> [1] Learning Transferable Visual Models From Natural Language Supervision. ICML 2021
>
> [2] Align before Fuse: Vision and Language Representation Learning with Momentum Distillation. NeurIPS 2021
>
> [3] BLIP: Bootstrapping Language-Image Pre-training for Unified Vision-Language Understanding and Generation. ICML 2022
>
> [4] BLIP-2: Bootstrapping Language-Image Pre-training with Frozen Image Encoders and Large Language Models. ICML 2023
>
> [5] Sigmoid Loss for Language Image Pre-Training. ICCV 2023

---

> > ### Author Rebuttal · Reviewer_5gbg · 2026-04-01
> >
> > I have no other questions, so I'll maintain my current rating for now. I will eventually consider the conclusions of other reviewers who gave negative scores.

---

### Official Review · Reviewer_4zX8 · 2026-03-11

**Soundness:** 3
**Presentation:** 3
**Significance:** 3
**Originality:** 3
**Overall Recommendation:** 4
**Confidence:** 3

**Summary:**

The paper presents APM, a framework designed for efficient Multi-modal Dataset Distillation (MDD). An important context considered by the study is the massive storage and computational bottleneck of current Trajectory Matching (TM) methods, which require storing numerous model checkpoints and performing costly bi-level optimization. Overall, a central concept presented by the paper is the derivation of analytic solutions (closed-form) for modal projectors under the InfoNCE loss to replace the inner-loop optimization. By matching these analytic parameters of real and synthetic data, the authors achieve up to 65× reduction in storage and 9.6× speedup in distillation time while maintaining superior retrieval performance.

**Compliance With Llm Reviewing Policy:**

Affirmed.

**Final Justification:**

I chose to keep my score.

**Key Questions For Authors:**

Is it possible to extend the APM framework to allow for partial updates of the encoder weights rather than just the projectors?
Have you tested the scalability of the analytic solution computation on extremely high-dimensional embeddings?

**Limitations:**

yes

**Strengths And Weaknesses:**

Strengths:

Exceptional Efficiency: The reduction in storage requirements and the significant speedup make large-scale multi-modal distillation accessible on standard hardware.

Theoretical Soundness: The authors provide rigorous mathematical derivations for both linear and non-linear projectors and address numerical instabilities like scale explosion.

Strong Generalization: The distilled datasets exhibit excellent cross-architecture generalization and perform well in zero-shot classification tasks.

Weaknesses:

Frozen Encoder Constraint: The current derivation focuses on optimizing projectors while keeping encoders frozen, which might limit performance if the backbone requires adaptation.

Matrix Inversion Scaling: While efficient, the \mathbit{O}(\mathbit{d}^\mathbf{3}) complexity of matrix inversion could become a factor as embedding dimensions \mathbit{d} continue to grow in future VLMs.

---

> ### Author Rebuttal · Authors · 2026-03-31
>
> We appreciate your detailed review and the recognition of our contributions.
>
> &nbsp;
>
> > **W1 & Q1: Frozen Encoder Constraint**
>
> A1: Currently, the APM framework cannot be applied to the weights of the encoder for two reasons:
>
> - A recent work on multi-modal dataset distillation, RepBlend [1], has shown that aligning the parameters of the modal projectors rather than the whole models can significantly improve the performance of MDD.
>
> - The image and text encoders have more complex architectures, such as Convolution and Transformer. To our knowledge, these architectures do not have analytic parameters currently.
>
> &nbsp;
>
> > **W2 & Q2: Matrix Inversion Scaling**
>
> A2: We truly understand your concerns about the matrix inversion. Empirically, we find that **the efficiency of APM is bounded by the number of synthetic pairs rather than the dimension of features**.
>
> ``[Experiment]`` To verify this, we evaluate the time and space overhead of APM on the Flickr dataset. Specifically, we vary the dimension of projectors from 256 to 2,048, and the number of synthetic pairs increases from 99 to 299. The results are shown below
>
> |Time (s/iter)|dim=256|dim=512|dim=1024|dim=2048|
> |-|-|-|-|-|
> |APM (99 pairs)|0.097|0.099|0.102|0.111|
>
> |Space (MB)|dim=256|dim=512|dim=1024|dim=2048|
> |-|-|-|-|-|
> |APM (99 pairs)|10710|10720|10790|10918|
>
> |Time (s/iter)|99 pairs|199 pairs|299 pairs|
> |-|-|-|-|
> |APM (dim=256)|0.097|0.175|**0.256**|
>
> |Space (MB)|99 pairs|199 pairs|299 pairs|
> |-|-|-|-|
> |APM (dim=256)|10710|19120|**27968**|
>
> ``[Reason]`` We can find that enlarging the dimension of modal projectors only slightly increases the time and space overhead of APM. The reasons are two-fold
>
> - The complexity of matrix inversion, e.g., $(V^{\top}V)^{-1}$, depends on the rank of $V^{\top}V$. As we mentioned in Lines 166-168, the largest number of synthetic pairs is 500, which is relatively smaller than the dimension of features. As a result, the rank of $V^{\top}V$ is bounded by the number of synthetic pairs.
>
> - For the real dataset with a large number of samples, we will calculate its matrix inversion **offline**, which will not affect the efficiency of APM during distillation.
>
> &nbsp;
>
> **Reference**
>
> [1] Beyond Modality Collapse: Representations Blending for Multimodal Dataset Distillation. NeurIPS 2025.

---

> > ### Author Rebuttal · Reviewer_4zX8 · 2026-04-03
> >
> > I have no other questions, so I'll maintain my current rating for now. I will eventually consider the conclusions of other reviewers who gave negative scores.

---

### Official Review · Reviewer_ZNKz · 2026-03-12

**Soundness:** 2
**Presentation:** 3
**Significance:** 2
**Originality:** 3
**Overall Recommendation:** 4
**Confidence:** 3

**Summary:**

This work proposed APM, a new approach for distilling multimodal datasets through first finding analytical solutions to the infoNCE loss and, second, synthesizing data via matching analytical parameters.

**Compliance With Llm Reviewing Policy:**

Affirmed.

**Final Justification:**

The rebuttal mostly clarified my concerns; my remaining concerns are with regard to the scope of the method itself and the usefulness of the task in practice, hence not a high score

**Key Questions For Authors:**

1. Is using non-linear projectors at all possible under the current methodology?

2.  In what sense and to what extent the real-world multi-modal datasets typically have low-rank structures is true, can authors elaborate?

3. Can the authors address the problem definition clarity?

**Limitations:**

1. If the non-linear projectors are more common in this line of work, what the authors propose only works for linear projectors. Is there a gap between linear and non-linear projectors in practice? If yes then this became a limitation.

2. Scope is somewhat narrow, CLIP-style linear projectors are reasonable for the first setting, but it is limiting in terms of generalization to modern end-to-end VLM training or architectures with nonlinear heads.

**Strengths And Weaknesses:**

Strength:

1.  Finding an analytical solution to the multimodel linear projectors that infoNCE is interesting.

2. Evaluation appears to be quite comprehensive, with many convincingly backing the contribution of the paper.

3. Good abstractions, and the authors extend the breadth beyond image-text.


Weakness:

1. Related work should be expanded to cover more works in dataset distillation and why those might not apply in multimodal setting.

2. Non-linear projector is discussed with some theoretical results(Proposition 3.2), but the evaluation only uses linear projectors.

3. The clarity of certain statements should be improved, for instance, in the sentence "Real-world multi-modal datasets typically have low-rank structures". Exactly what is low rank, the data themself or the interactions between multimodal data.

4. The authors introduced an optimization objective in equation 2 which TM attempts to solve, the implication of APM with regards to this optimization problem is unclear.

5.  I found the problem setup somewhat unclear. In particular, it is difficult to tell if the method is solving the projector-learning objective in closed form or optimizing a matching loss for dataset distillation.

6. In equation 1, u and v are normalized. But normalization does not appear in the definition of H,U and V or in Proposition 3.1.

---

> ### Author Rebuttal · Authors · 2026-03-31
>
> Thanks for the detailed and helpful comments. We reply to the comments in detail. Hope to address your concerns.
>
> &nbsp;
>
> > **W1: Related work and why those might not apply in multimodal setting**
>
> A1: In addition to TM and APM, other DD methods can also be used in multimodal settings, such as gradient matching (GM). However, these methods are less efficient because they rely on bi-level optimization. For example, GM first computes gradients on the real and synthetic datasets and then minimizes their difference to update the synthetic data. We will clarify this discussion in the revision.
>
> &nbsp;
>
> > **W2 & Q1: Non-linear projector experiments**
>
> A2:
> A1: Both linear and non-linear projectors can be used in our method. To verify this, we conduct additional experiments on Flickr and COCO using ``Tanh`` as the non-linear activation, and report the results below.
>
> |Flickr Pairs|Method|IR@1|IR@5|IR@10|TR@1|TR@5|TR@10|
> |-|-|-|-|-|-|-|-|
> |100|APM|12.8|34.2|47.1|17.8|43.0|57.2|
> ||APM-Tanh|10.6|30.5|42.2|14.7|37.8|51.5|
> |200|APM|14.6|38.5|52.0|18.9|47.8|62.2|
> ||APM-Tanh|12.0|33.5|45.2|15.6|44.8|57.5|
> |500|APM|17.5|43.5|56.8|21.6|52.7|66.4|
> ||APM-Tanh|15.3|40.1|49.3|20.5|48.9|63.0|
>
> |COCO Pairs|Method|IR@1|IR@5|IR@10|TR@1|TR@5|TR@10|
> |-|-|-|-|-|-|-|-|
> |100|APM|4.7|16.2|25.8|6.2|20.0|31.1|
> ||APM-Tanh|3.9|12.8|21.5|4.7|16.6|27.5|
> |200|APM|6.1|19.6|30.4|7.7|23.6|35.3|
> ||APM-Tanh|5.5|17.4|26.5|6.0|19.8|31.5|
> |500|APM|7.1|21.8|33.3|8.0|24.3|37.1|
> ||APM-Tanh|6.3|19.5|31.0|6.6|21.8|31.3|
>
> We observe that matching analytic parameters for non-linear projectors performs worse than for linear projectors. A likely reason is numerical instability: the analytic parameters require the inverse activation, i.e., $\sigma^{-1}(V(V^{\top}V)^{-1})$. For Tanh activation, **the inverse is only defined on (-1, 1)**, while $V(V^{\top}V)^{-1}$ is unbounded. Truncating the values may therefore introduce instability.
>
> Our main goal is to improve the efficiency of MDD while preserving performance. Our theory of the non-linear projection operator is mainly a feasibility analysis. We will add these observations and the discussion in the revision.
>
> &nbsp;
>
> > **W3 & Q2: Statement of low-rank structures in multi-modal datasets**
>
> A3: Our claim is that the image and text embeddings exhibit low-rank structure. The analytic parameter of the modal projector uses matrix inversion to improve the rank of data embeddings.
>
> This is also one reason why APM works well: it matches the high-rank structure of the real and synthetic datasets, thus improving the data entropy of synthetic pairs. Figure 3 supports this observation.
>
> &nbsp;
>
> > **W4: Optimization objective of TM and APM**
>
> A4: The optimization objective of APM has a similar motivation to TM: Aligning the parameters of modal projectors trained on the real and synthetic datasets.
>
> Their differences are two-fold:
>
> - RepBlend is **multi-step** matching, while APM is **one-step** matching.
>     - RepBlend records the training trajectory on real dataset and enforces model trained on the synthetic dataset has a similar trajectory.
>     - APM only matches the analytic parameters between the real and synthetic datasets.
>
> - RepBlend uses **bi-level** optimization, while APM is **single-level** optimization.
>     - In the inner-loop, RepBlend updates model parameters on synthetic dataset. In the outer-loop, RepBlend minimizes the differences between teacher and student trajectories.
>     - APM replaces the inner-loop with analytic parameters, which can be directly calculated in a forward pass.
>
> &nbsp;
>
> > **W5 & Q3: Problem Setup**
>
> A5: In Equation 2, we formulate multimodal dataset distillation (MDD) as a bi-level optimization problem:
>
> - The inner loop trains a model on the synthetic dataset until convergence;
> - The outer loop updates the synthetic pairs by minimizing the InfoNCE loss on the real dataset.
>
> Directly solving this problem is time- and resource-intensive. Therefore, the problem is simplified by minimizing the difference between models trained on real and synthetic datasets. Empirically, the information of the teacher model can be cached offline, such as gradients (GM), trajectories (TM), or analytic parameters (APM), which reduces online computation.
>
> &nbsp;
>
> > **W6: Normalization of U and V**
>
> A6: The normalization of $U$ and $V$ cannot be absorbed into the analytic parameters.
>
> Let $U$ and $V$ be the unnormalized image and text embeddings, and define $U\_2=U/||U||\_2$, $V\_2=V/||V||\_2$. Based on Proposition 3.2, we can obtain the analytic parameters of $U\_2$ and $V\_2$.
> However, we cannot recover $U$ and $V$ from $U\_2$ and $V\_2$ since infinitely many $U$ and $V$ satisfy the same normalized form.

---

> > ### Author Rebuttal · Reviewer_ZNKz · 2026-04-03
> >
> > Thank you for addressing my questions. I will give a positive rating.

---

> > > ### Author Response · Authors · 2026-04-04
> > >
> > > Dear Reviewer ZNKz,
> > >
> > > We sincerely appreciate your helpful suggestions and are glad to know that our revision has addressed all your concerns.
> > >
> > > Since your current comments appear positive, we would be truly grateful if you could consider updating the **Overall Recommendation**.
> > >
> > > Thank you again for your time and support.
> > >
> > > Authors of Submission 4699

---

### Official Review · Reviewer_mTLX · 2026-03-13

**Soundness:** 3
**Presentation:** 2
**Significance:** 3
**Originality:** 3
**Overall Recommendation:** 4
**Confidence:** 4

**Summary:**

This paper proposes efficient MDD method that matches only analytic parameters of projection layers.
Analytic Parameter Matching (APM) deals with Embedding Shift, Scale Explosion, and Matrix Inversion.
And also the paper introduces knowledge distillation mechanism at the evaluation stage.
The experimental results show that proposed method achieves good performance and the synthetic dataset has high data entropy with large diversity.

**Compliance With Llm Reviewing Policy:**

Affirmed.

**Final Justification:**

The paper introduces a efficient multi-modal DD paradigm that matches parameters of projection layers only.
The authors well addressed my concerns, thus I raise my score accordingly.

**Key Questions For Authors:**

1. Could you provide non-linear case experiments?
2. What is the main difference between RepBlend and APM? and why RepBlend outperforms APM (Flickr, 500Pairs, TR)?
3. Could you provide an ablation study w.r.t L_KD?
4. Could you explain more details about the implementation of Table 6?
5. According to W3, could you compare the evaluation time with other methods?

**Limitations:**

yes

**Strengths And Weaknesses:**

**Strengths**
1. The paper is well-organized and easy to follow.
2. The paper presents theoretically well-motivated problem and the proposed method properly deals with corresponding problem.
3. Performance improvements are noticeable on MS-COCO dataset.

**Weaknesses**
1. The method generalizes to non-linear case but experiments showcase only linear case.
2. Performance degrades in some cases on Flickr without sufficient explanation.
3. The evaluation requires online teacher forward process which leads to additional computation.
4. Table 6, it is missing how ES/SE/MI are separately ablated.

-minor comment:
maybe, L_MCL should be replaced L_NCE if I am correct.

---

> ### Author Rebuttal · Authors · 2026-03-31
>
> We are grateful for your constructive comments.
>
> &nbsp;
>
> > **W1 & Q1: Non-linear case experiments**
>
> A1: Both linear and non-linear projectors can be used in our method. To verify this, we conduct additional experiments on Flickr and COCO using ``Tanh`` as the non-linear activation, and report the results below.
>
> |Flickr Pairs|Method|IR@1|IR@5|IR@10|TR@1|TR@5|TR@10|
> |-|-|-|-|-|-|-|-|
> |100|APM|12.8|34.2|47.1|17.8|43.0|57.2|
> ||APM-Tanh|10.6|30.5|42.2|14.7|37.8|51.5|
> |200|APM|14.6|38.5|52.0|18.9|47.8|62.2|
> ||APM-Tanh|12.0|33.5|45.2|15.6|44.8|57.5|
> |500|APM|17.5|43.5|56.8|21.6|52.7|66.4|
> ||APM-Tanh|15.3|40.1|49.3|20.5|48.9|63.0|
>
> |COCO Pairs|Method|IR@1|IR@5|IR@10|TR@1|TR@5|TR@10|
> |-|-|-|-|-|-|-|-|
> |100|APM|4.7|16.2|25.8|6.2|20.0|31.1|
> ||APM-Tanh|3.9|12.8|21.5|4.7|16.6|27.5|
> |200|APM|6.1|19.6|30.4|7.7|23.6|35.3|
> ||APM-Tanh|5.5|17.4|26.5|6.0|19.8|31.5|
> |500|APM|7.1|21.8|33.3|8.0|24.3|37.1|
> ||APM-Tanh|6.3|19.5|31.0|6.6|21.8|31.3|
>
> We observe that matching analytic parameters for non-linear projectors performs worse than for linear projectors. A likely reason is numerical instability: the analytic parameters require the inverse activation, i.e., $\sigma^{-1}(V(V^{\top}V)^{-1})$. For Tanh activation, **the inverse is only defined on (-1, 1)**, while $V(V^{\top}V)^{-1}$ is unbounded. Truncating the values may therefore introduce instability.
>
> We will add these observations and the discussion in the revision.
>
> &nbsp;
>
> > **W2: Performance between RepBlend and APM on Flickr**
>
> A2: APM outperforms RepBlend across three datasets and settings, except for the Flickr dataset, text retrieval (TR), 500 pairs. We attribute this degradation to the teacher models.
>
> RepBlend uses a heavy projector to improve the performance of its teacher model and outperforms APM on the TR setting. As both methods use similarity mining, the teacher model of RepBlend can provide a more accurate similarity matrix as pseudo-supervision.
>
> &nbsp;
>
> > **W3 & Q5: The evaluation requires online teacher forward process**
>
> A3: The additional computation arising from the teacher forward is acceptable for two reasons:
>
> - In the evaluation phase, we only call the teacher model **once** to calculate the similarity matrix of the synthetic dataset. As a result, the additional computation should be amortized by each epoch in the evaluation phase.
>
> - The similarity mining technique is widely used in multi-modal dataset distillation. Therefore, the evaluation time of APM is the same as other models.
>
> &nbsp;
>
>
> > **W4 & Q4: Implementation of Table 6 (Ablation)**
>
> A4: In Table 6, we aim to verify the effectiveness of the three normalizations in APM, including ES/SE/MI.
>
> We separately ablate these normalizations from $\mathcal{L}_{APM}$ rather than sequentially removing them. Specifically, in Table 6,
>
> - w/o ES indicates that the data embeddings are no longer zero-meaned, e.g., replacing $H_I - \mu_I$ with $H_I$.
>
> - w/o SE represents that the data covariance is not normalized by the number of samples, e.g., removing $\frac{1}{|\mathcal{D}|}$.
>
> - w/o MI means that the stability term $\alpha I$ is removed when calculating the matrix inversion.
>
> &nbsp;
>
> > **Q2: Difference between RepBlend and APM**
>
> A5: Both RepBlend and APM aim to match the parameters of modal projectors trained on the real and synthetic datasets. Their differences are two-fold:
>
> - RepBlend is **multi-step** matching, while APM is **one-step** matching, **reducing storage cost**.
>     - RepBlend records the training trajectory on real dataset and enforces model trained on synthetic dataset has a similar trajectory. The loss function is defined as $\mathcal{L}\_{Rep}=\sum_{i} ||\theta_i - \hat{\theta}_i||_F^2$.
>     - APM only matches the analytic parameters between the real and synthetic datasets. The loss function is defined as $\mathcal{L}\_{APM}=||\theta^* - \hat{\theta}^*||_F^2$.
>
> - RepBlend uses **bi-level** optimization, while APM is **single-level** optimization, **reducing computational cost**.
>     - In inner loop, RepBlend updates model parameters on synthetic dataset. In outer loop, RepBlend minimizes the differences between teacher and student trajectories.
>     - APM replaces the inner-loop with analytic parameters, which can be directly calculated in a forward pass.
>
> &nbsp;
>
>
> > **Q3: Ablation study on L_KD**
>
> A6: We report the results for three methods without teacher model supervision. The results are shown below
>
> |Flickr-100|IR@1|IR@5|IR@10|TR@1|TR@5|TR@10|
> |-|-|-|-|-|-|-|
> |Random|1.0|4.0|6.5|1.3|5.9|10.1|
> |RepBlend|3.7|12.5|18.8|6.3|24.7|27.0|
> |APM|4.5|15.2|23.9|8.6|27.1|37.9|
>
> We can observe that $\mathcal{L}_{KD}$ is essential for the performance of MDD. Despite the performance degradation of APM, it consistently outperforms RepBlend and Random.
>
> &nbsp;
>
> > **Minor: L_MCL and L_NCE**
>
> A7: Thanks for pointing out this issue. We will replace the $\mathcal{L}\_{MCL}$ with $\mathcal{L}\_{NCE}$ in the revision.

---

> > ### Author Rebuttal · Reviewer_mTLX · 2026-04-01
> >
> > Thank you for your detailed rebuttal. I have a few follow-up questions:
> >
> > 1. Regarding W2, could you provide additional results using a heavy projector, which is adopted in RepBlend?
> >
> > 2. The computational complexity at evaluation time may differ. As noted in the paper (Similarity Mining), the implementation differs from LoRS and includes an additional SVD step. Furthermore, are any data augmentation strategies applied? If so, the teacher model would need to perform forward passes per batch to compute the exact similarity matrix, which could further affect evaluation cost. A precise comparison of evaluation time would help clarify this concern.
> >
> > Please correct me if I misunderstood.

---

> > > ### Author Response · Authors · 2026-04-02
> > >
> > > Thanks for your quick follow-up.
> > >
> > > &nbsp;
> > >
> > > > **Q1: Details in the projector**
> > >
> > > A1: The heavy projectors contain more than two layers of MLP. Below, we discuss two cases of applying heavy projectors to APM
> > >
> > > - **Distillation with heavy projectors**. The projected data embeddings can be defined as $U=\sigma(H\_I W\_I^1)W_I^2$ and $V=\sigma(H\_T W\_T^1)W\_T^2$. Based on Equation 17, we have $U=V(V^{\top}V)^{-1}=\sigma(H\_T W\_T^1)W\_T^2$. We can observe that the analytic parameters of $W\_T^1$ depend on $W\_T^2$, and vice versa. Therefore, it is infeasible to give the analytic parameters of the heavy projectors.
> > >
> > > - **Evaluation with heavy projectors**. While heavy projectors make analytic distillation infeasible, they can be used as more expressive evaluation networks. We re-evaluate the synthetic datasets (distilled by APM) using the projectors in RepBlend. The results are shown below. We can see that APM outperforms RepBlend and exhibits strong cross-architecture generalization.
> > >
> > > |Flickr-500|IR@1|IR@5|IR@10|TR@1|TR@5|TR@10|
> > > |-|-|-|-|-|-|-|
> > > |RepBlend|17.0|42.5|55.9|22.5|53.2|**66.7**|
> > > |APM|**17.8**|**43.3**|**56.4**|**23.1**|**53.5**|**66.7**|
> > >
> > > &nbsp;
> > >
> > > > **Q2: Details in the evaluation phase**
> > >
> > > A2: For methods using similarity mining, including LoRS, RepBlend, and APM, the same function [``evaluate_synset_with_similarity``](https://github.com/silicx/LoRS_Distill/blob/8a712afdb30ed483199bd0f00cc73755e0817fe8/src/epoch.py#L260) is used for evaluation.
> > >
> > > **Data augmentation is not used** in this evaluation function, so the teacher model does not need to perform forward passes in each epoch.
> > >
> > > We clarify the differences between LoRS and APM in detail
> > >
> > > - LoRS initializes two low-rank matrices $L$ and $R$ and optimizes them during distillation, which introduces additional computations. After distillation, LoRS calculates the similarity matrix, $S=wI+\frac{\alpha}{r}LR^{\top}$.
> > >
> > > - APM does not optimize the similarity matrix during distillation. After distillation, APM calls the teacher model to generate a similarity matrix, $S=\mathcal{M}(\hat{X}, \hat{K})$. Additional SVD can be used to reduce the size of $S$.
> > >
> > > - During evaluation, the similarity matrix is fixed for both LoRS and APM. In this case, the calculation of the similarity matrix should be treated as a **pre-processing step**.
> > >
> > > We truly understand your concerns about the complexity of evaluation. Below, we report the time overhead of APM and LoRS in the pre-processing and evaluation stages. We can observe that APM and LoRS have similar evaluation times.
> > >
> > > |500 Pairs|Pre-processing|Evaluation (100 epochs)|
> > > |-|-|-|
> > > |LoRS|0.08s|25.6s|
> > > |APM|0.01s|25.7s|

---

### Decision · Program_Chairs · 2026-04-30

**Decision:**

Accept (regular)

**Comment:**

This submission received consistently positive, though not enthusiastic, assessments: all four reviewers recommended weak accept. The reviewers broadly agreed that the paper’s main contribution, replacing trajectory-based multimodal dataset distillation with analytic parameter matching for modal projectors, is novel and technically meaningful, with clear efficiency benefits and generally solid empirical support across image-text and audio-text settings. Several reviewers also viewed the paper as theoretically motivated and appreciated evidence of cross-architecture and zero-shot generalization. At the same time, the main limitations were also consistent: the method’s scope is restricted to projector matching with frozen encoders, practical relevance beyond linear or CLIP-style heads remains uncertain, and the original submission left some concerns about clarity, related work, ablations, nonlinear settings, and evaluation-time cost. The rebuttal addressed a substantial portion of these issues by clarifying the problem setup, adding nonlinear and efficiency evidence, and explaining ablations and comparison points. However, it did not fully remove concerns about scope and ultimate impact in broader VLM settings.